# Gut Digestive Function and Microbiome after Correction of Experimental Dysbiosis in Rats by Indigenous Bifidobacteria

**DOI:** 10.3390/microorganisms9030522

**Published:** 2021-03-04

**Authors:** Lyudmila V. Gromova, Elena I. Ermolenko, Anastasiya L. Sepp, Yulia V. Dmitrieva, Anna S. Alekseeva, Nadezhda S. Lavrenova, Mariya P. Kotyleva, Tatyana A. Kramskaya, Alena B. Karaseva, Alexandr N. Suvorov, Andrey A. Gruzdkov

**Affiliations:** 1Pavlov Institute of Physiology, Russian Academy of Sciences, 199034 Saint-Petersburg, Russia; seppal@infran.ru (A.L.S.); dmitrievayv@infran.ru (Y.V.D.); alekseevaas@infran.ru (A.S.A.); gruzdkovaa@infran.ru (A.A.G.); 2Department of Molecular Microbiology, Institute of Experimental Medicine, 197376 Saint-Petersburg, Russia; Lermolenko1@yandex.ru (E.I.E.); nadezhda.lavrenova.vrn@gmail.com (N.S.L.); mariha.lenivaya@mail.ru (M.P.K.); tatyana.kramskaya@gmail.com (T.A.K.); tarno@list.ru (A.B.K.); alexander_suvorov1@hotmail.com (A.N.S.); 3Department of Medical Microbiology, North-Western State Medical University Named after I.I. Mechnikov, Ministry of Health of the Russian Federation, 195067 Saint-Petersburg, Russia; 4Department of Fundamental Problems of Medicine and Medical Technologies, Saint-Petersburg State University, 199034 Saint-Petersburg, Russia

**Keywords:** autoprobiotics, intestinal digestive enzymes, microbiome

## Abstract

In recent years, great interest has arisen in the use of autoprobiotics (indigenous bacteria isolated from the organism and introduced into the same organism after growing). This study aimed to evaluate the effects of indigenous bifidobacteria on intestinal microbiota and digestive enzymes in a rat model of antibiotic-associated dysbiosis. Our results showed that indigenous bifidobacteria (the Bf group) accelerate the disappearance of dyspeptic symptoms in rats and prevent an increase in chyme mass in the upper intestine compared to the group without autoprobiotics (the C1 group), but significantly increase the mass of chyme in the colon compared to the C1 group and the control group (healthy animals). In the Bf group in the gut microbiota, the content of opportunistic bacteria (*Proteus* spp., enteropathogenic *Escherichia coli*) decreased, and the content of some beneficial bacteria (*Bifidobacterium* spp., *Dorea* spp., *Blautia* spp., the genus *Ruminococcus*, *Prevotella*, *Oscillospira*) changed compared to the control group. Unlike the C1 group, in the Bf group there was no decrease in the specific activities of maltase and alkaline phosphatase in the mucosa of the upper intestine, but the specific activity of maltase was decreased in the colon chyme compared to the control and C1 groups. In the Bf group, the specific activity of aminopeptidase N was reduced in the duodenum mucosa and the colon chyme compared to the control group. We concluded that indigenous bifidobacteria can protect the microbiota and intestinal digestive enzymes in the intestine from disorders caused by dysbiosis; however, there may be impaired motor function of the colon.

## 1. Introduction

Human and animal health largely depends on the interaction between the intestinal epithelium and its microbiota. The main role in the implementation of this interaction belongs to the large intestine, which contains the largest number of microorganisms involved in the breakdown of undigested food and endogenous components (mucus, enzymes) [1,2,3,4]. At the same time, the intestinal microbiota is involved in maintaining the protective barrier of the intestine as one of the mechanisms of this barrier, which is able to control the development and maintenance of the intestinal immune system [1,3,5].

Bacteria belonging to the genus Bifidobacterium are among the main constituents of the intestinal microbiota in humans and animals [4,6,7,8,9]. Despite the small content of these bacteria in the intestine of adult organisms in a healthy state (in humans, bifidobacteria make up 3–6% of the total number of bacteria in feces), their presence has a beneficial effect on the organism [4,7,8,9,10]. They participate in the fermentation of food and endogenous carbohydrates (sucrose, galactose, fructose, lactose, and some oligosaccharides present in human milk) and suppress the growth of pathogenic microorganisms by the production of organic acids (acetic, lactic) and bacteriocins by competing for receptors on the intestinal epithelium, on which fixation of pathogenic microorganisms can occur, and by deconjugation of bile acids [9,10].

At the same time, bifidobacteria participate in the modulation of the immune system (in particular, by changing the production of mucus by goblet cells and by the degradation of protective glycans on mucins) [9] and restoration of the altered microbiota, significantly affecting its composition [8,9,10]. Due to these effects, probiotic bifidobacteria alone or in combination with other bacteria are often used for the prevention and correction of dysbiosis.

In recent years, autoprobiotics—indigenous bacteria grown in laboratory conditions—have attracted great interest, as means for the correction of intestinal dysbiosis [11]. This approach takes into account the fact that, in comparison with probiotic bacteria, representatives of own microbiota have better immunological tolerance, and do not show pronounced antagonistic activity in relation to obligate members of intestinal microbiota [11] and can persist for a long time in the body, due to better adaptation to the conditions of existence in it.

Despite significant advances achieved in recent years into these topics, the action of individual probiotics and, especially, autoprobiotics in relation to the correction of the microbiota and the restoration of the digestive and protective functions of the intestine in most cases has not yet been sufficiently studied. Knowledge of these issues is especially important when using auto-probiotics for personalized therapy.

The purpose of this work was to study the effects of autoprobiotic bifidobacteria on microbiota and intestinal membrane enzymes involved in digestion and maintaining homeostasis using a rat model of antibiotic-associated dysbiosis.

## 2. Materials and Methods

### 2.1. Animals

Wistar rats (males, 200–250 g, 6–7 weeks old), obtained from the Animal Breeding Center, (Rappolovo, Russia) were kept in separate cages under constant conditions: Room temperature (18–22 °C), 12-h light/dark cycle, a noise level of no more than 85 dB, humidity of 50–60%. They were given free access to water and standard feed (complete ration for laboratory rats and mice PK-120 sh. 1492, GOST R 50258-92, granules of 14 mm in diameter, Russia). The study carried out in strict accordance with the necessary ethical requirements and in accordance with the principles of humane treatment of animals (European Community No. 86/609 EC). The study was approved by the local ethics committee of the Institute of Experimental Medicine.

### 2.2. Rat Model of Antibiotic-Associated Dysbiosis

Experimental intestinal dysbiosis was induced in rats via daily intragastrical administration (by means of metal tip) of ampicillin ^®^ (Orgenica, Moscow, Russia) at a dose of 75 mg/kg and metronidazole ^®^ (Nycomed, Zürich, Switzerland) at a dose of 50 mg/kg for three days [12,13,14].

### 2.3. Autoprobiotic Strains of Bifidobacteria

For the preparation of auto-probiotic bifidobacteria, the feces were taken from healthy rats seven days before the administration of antibiotics. Individual clones of *Bifidobacterium* spp. were isolated from the Blauroca broth (Nutrient medium, Saint-Petersburg, Russia) after incubation of fecal samples for 48 h at 37 °C anaerobically. Five typical colonies grown at the bottom of the tubes were subcultured onto fresh culture medium to obtain a pure culture. Individual clones of bifidiobacteria were identified using microscopy and employing a RT-PCR set of primers encoding for 16S rRNAsequence (Table 1). They were then harvested by centrifugation at 3000× *g* for 10 min, suspended in sterile PBS (phosphate buffer saline) at a concentration of 5.5 × 10^8^ CFU/mL and stored at −20 °C until use. PBS composition: 8.00 g/L NaCl, 0.20 g/L KCl, 1.44 g/L Na_2_HPO_4_, 0.24 g/L KH_2_PO_4_ (pH 7.4).

In this study, we did not intend to identify specific species of bifidobacteria.

### 2.4. Design of the Study

The rats were divided into three groups (n = 10–15 animals in each group): Control group C0 (healthy animals), group C1, and group Bf. The rats in the groups Bf and C1, after application of antibiotics for 3 days, received then for 5 days autoprobiotic bifidobacteria and PBS (instead of auto-probiotic), respectively. Rats in the control group C0 received water instead of antibiotics and PBS instead of auto-probiotic at the same time (Table 2).

It is important to note that the rats of the Bf group throughout the entire period of the experiment (including the period with the introduction of autoprobiotic bifidobacteria) were kept in separate cells in order to avoid the exchange of microbiota with other animals of this group. In addition, the diet of rats in our experiments did not contain bifidobacteria. These facts confirm that in our experiments, the rats of group Bf received only indigenous bifidobacteria. Fecal samples harvested on days 0 and 9 of experiments from all animals were used for microbiota study by qPCR and by metagenome analysis.

After the necropsy examination (ninth day of the experiments), the state of the intestine was assessed macroscopically. The small intestine, the large intestine, and the cecum were weighed separately. The samples of mucosa and chyme were taken from various parts of the small intestine (duodenum, jejunum, ileum) and from the colon. They were stored at −80 °C and frozen to determine the activity of digestive enzymes of the intestinal epithelium and the chyme.

### 2.5. Microbiome Study

The fecal samples were analyzed by 16S rRNA gene-based metagenomics analysis. Changes in the gut microbiota content were investigated by performing 16S rRNA gene-based metagenome analysis using a previously described approach [11]. Changes in the gut microbiome content were studied by quantitative PCR (qPCR) and 16S rRNA gene-based metagenomic analysis. Fecal samples were collected on day 9.

qPCR system for microbiota analysis («Colonoflor» AlphaLab, Saint-Petersburg, Russia) was used to characterize the main gut bacterial groups by quantitative (real-time) PCR. PCR amplification and detection were performed with Mini Opticon (MJMini, BIORAD, Hercules, CA, USA).

The necessary temperature regime of the reaction was established (DNA denaturation −95 °C for 3 min, then 58 cycles under the following conditions: Denaturation for 10 s at 95 °C, annealing of primers 30 s at 55 °C, and the last stage-DNA chain elongation at 72 °C for 60 s). The probes’ DNA were labeled by HEX or FAM dyes. Analysis of PCR results in real time was carried out on the Bio-Rad device. Quantitative evaluation of the analyzed bacteria in the sample was noted on the starting point of the exponential growth curve and the program determined the cycle of amplification that corresponded to the point (threshold cycle C (t)). A special table (test system “Colonoflor”) was used to determine the number of CFU of bacteria contained in a test sample of feces. It could be used to correlate the value of the obtained threshold cycle C (t) with the number of CFU of bacteria in the sample.

#### 2.5.1. Metagenome Analysis

DNA from feces for 16S rRNA gene-based metagenome analysis was isolated using the QIAamp DNA Stool Mini Kit (Qiagen N.V., Venlo, Netherlands) following the manufacturer’s protocol. Samples were incubated in the lysis buffer at 90 °C for 10 min for optimal bacterial lysis. For microbiome sequencing, DNA libraries were prepared using the Illumina Nextera Sample Preparation Kit with DNA primers corresponding to V3–V4 regions of the 16S rRNA. Illumina MiSeq was used for sequencing the libraries (Table 3).

#### 2.5.2. OTU Generation

Simultaneous clustering of reads of all analyzed samples yielded 7782 OTUs. Most of the OTUs were present in both a low number of samples and low absolute amounts, which is a common feature of gut microbiome samples. Most of these small groups of reads failed to match any sequences from database Greengene 13.5 used for annotation. The number of annotated OTUs was 534 with the percent of unclassified reads being −22.4%. OTUs present in less than 5% of samples were discarded for noise filtering.

Fastqc (http://www.bioinformatics.babraham.ac.uk/projects/fastqc (accessed on 20 November 2020)) was used to evaluate the quality of raw reads. CD-HIT-OTU-Miseq was used for OTU retrieval. CD-HIT-OTU-Miseq allowed us to retrieve OTU from paired-end reads without merging paired sequences. This was achieved by matching the clustering results for R1 and R2 reads. CD-HITOTU-Miseq could use only high-quality regions of reads for clustering. Clustering was performed using the following parameters: Lengths of high-quality regions of R1 and R2 reads of 200 and 180 bp, respectively, 97% read similarity for clustering cutoff and 0.00001 for abundance cutoff. OTUs were annotated using Greengenes database version 13.5 [15]. In total, there were 9,754,220 reads. After filtering by quality of the reads, selected the length for R1 and R2 reads, 1,558,276 (15.9%) remained. The parameters for the Trimomatic program that filters the reads were MINLEN 200 and 180 SLIDINGWINDOW:4:20 LEADING:3 TRAILING:3 MINLEN:{MINLEN} MAXINFO:80:0.5.

Vectors for PCA analysis corresponded to OTU abundances filtered for noise and normalized for total OTU counts for each sample. The noise filtering cutoff was increased for PCA to discard OTUs present in less than 25% of samples.

Detailed results and conditions of sequencing are presented in the Appendix A section.

### 2.6. Biochemical Analysis

Maltase (ML, EC 3.2.1.20), alkaline phosphatase (AP, EC 3.1.3.1), and aminopeptidase N (AMN, EC 3.4.11.2,) activities were determined in homogenates of chyme from different segments of the intestine using the methods described earlier [14,16]. Chyme samples were obtained from the small intestine, as well as from the colon. For this purpose, the lumen of each section of the intestine was washed with cold Ringer’s solution (pH 7.1–7.4), 30 mL. Specific activity of every enzyme was calculated as µmol/min per 1 g wet weight of mucosa or chyme collected from the intestinal segment.

### 2.7. Statistical Analysis

Statistical analysis was performed using the software package Statistica 8.0. (StatSoft Inc., Tulsa, OK, USA). Differences between the groups were analyzed using Student’s *t*-test, Kruskal–Wallis test, ANOVA with post-hoc HSD test for unequal n, and MANOVA analyses in Python stats model; *p* < 0.05 was considered as significant.

## 3. Results

### 3.1. Health Status and Body Weight of Rats

When antibiotics were administered for 3 days in groups C1 and Bf, dyspepsia symptoms (which were expressed by the soft consistency of feces) was observed in animals. On the third day after the start of antibiotic administration, the number of animals with the soft consistency of feces was as follows: In the C1 group—4/15, and in the Bf group—3/10. In absence of autoprobiotics, after the withdrawal of antibiotics (the C1 group), the symptoms of dyspepsia in rats disappeared on the eighth day of the experiments, whereas in the presence of indigenous bifidobacteria (the Bf group), these symptoms disappeared already on the sixth day of the experiments.

On the fourth day after the start of antibiotic administration, the body weight of rats in the C1 group did not change, and in the Bf group it slightly decreased as compared to the first day. However, in the control group C0, the body weight of rats increased significantly from the first to the fourth days (*p* < 0.05) (Figure 1A). Subsequently, from the fourth to the eighth day of experiments in the Bf group and in the C1 group, the rats showed an increase in body weight (*p* < 0.01 and *p* < 0.003 for groups Bf and C1, respectively). This increase in body weight in groups Bf and C1 did not differ from the weight gain in the control group C0 for the same period.

Food intake by rats in the control group C0 tended to decrease from 1 to 7 days of the experiments (Figure 1B). In groups Bf and C1, food consumption by rats from the first to third days of the experiments (the period of administration of antibiotics in these groups) was reduced in comparison with the control group C0 (significantly in the C1 group on the third day (*p* < 0.05). On the fifth and on the seventh days of the experiments (the period of the introduction of the autoprobiotic or in its absence), the food consumption by the rats in the Bf and C1 groups increased in comparison with the first and third days of the experiments (for each of the groups Bf and C1 on the fifth and the seventh days as compared to third day: *p* < 0.05 and *p* < 0.003, respectively), as well as in comparison with the control group C0 on the seventh day of experiments (for each of the groups Bf and C1: *p* < 0.003).

### 3.2. Mass of Mucosa and Chyme in Intestinal Segments

At the end of the experiments, the rats of the C1 group showed a tendency to a decrease in the mass of mucosa in the colon, while in the Bf group rats, this indicator was significantly reduced (by 28.6%, *p* < 0.05) compared to the C0 control group (Figure 2A).

Significant differences were found in the distribution of the chyme mass along the intestine in groups C0, C1, and Bf at the end of the experiments. Thus, in the C1 group, compared with the C0 group, there was a tendency to increase in the mass of chyme in the duodenum, as well as in the proximal and distal parts of the jejunum. At the same time, in group Bf, the mass of the chyme in the duodenum and in the proximal jejunum was reduced (*p* < 0.05) in comparison with group C1, and did not differ from the corresponding values in the C0 group. It is noteworthy that in group Bf, the mass of chyme in the colon was significantly increased compared with groups C0 and C1 (approximately 3 times, *p* < 0.01) (Figure 2B).

At the end of the experiments, an increase in the ratio of the weight of the cecum (including its contents) to the body weight of animal was also found in rats in the groups C1 and Bf compared to the control C0 (Figure 3).

### 3.3. Microbiota Study by qPCR

As shown in Figure 4A, the content of *Bifidobacterium* spp. in fecal samples from rats in group Bf was higher than in the control C0 and group C1 (*p* < 0.05). At the same time, the content of *Proteus* spp. and enteropathogenic *Escherichia coli* in the fecal microbiota in group Bf was reduced compared to group C1 (*p* < 0.05) and did not differ significantly from control C0 (Figure 4B,C).

### 3.4. Metagenome Analysis of Microbiota

The analysis of the metagenome revealed that in group Bf, the relative abundance of bacteria belonging to the genera *Ruminococcus, Prevotella,* and *Oscillospira* was reduced, and in the genera *Bifidobacterium* spp., *Blautia* spp., and *Dorea* spp., it increased in comparison with the control C0 (*p* < 0.05 or *p* < 0.01) (Figure 5A–G). In group C1, a decrease in the population of lactobacilli and bifidobacteria was noted at the genera level. (Figure 5A–C). At the same time, in the case of some bacteria (*Bifidobacterium* spp., *Lactobacillus* spp., *Blautia* spp., *Dorea* spp.) in group Bf, their relative abundance was noticeably higher than in group C1 (*p* < 0.05 or *p* < 0.01).

When examining alpha diversity, no significant differences were found between groups C0, C1, and Bf. However, it should be noted that biodiversity was highest in group C0 and lowest in group C1 (Appendix A).

In addition, we compared the differences in the abundance of some genera (*Bacteroides, Bifidobacterium, Blautia, Dorea, Lactobacillus, Oscillospira, Prevotella, Ruminococcus*) in different groups using subsequent MANOVA analysis in a Python statistical model. Statistically significant differences were revealed between groups C1 and C0 (*p* < 0.019, Appendix A) and between C1 and Bf (*p* < 0.02, Appendix A). It is important to note that when comparing groups C0 and Bf, no significant difference was found (Appendix A).

Metagenome composition of animal groups obtained after principal component analysis (PCA) of sample OTU compositions is presented in Figure 6. As can be seen, the metagenome compositions of animals from group C1 and from groups C0 and Bf are grouped mainly in opposite parts of this figure.

Finally, to analyze the differences in the relative abundance of these bacterial genera (*Bacteroides, Bifidobacterium, Blautia, Dorea, Lactobacillus, Oscillospira, Prevotella, Ruminococcus*) and the correlations between them, we used an approach when all three groups were analyzed together. However, the correlations between the representation of the individual genera of bacteria was not established (Appendix A).

### 3.5. The Activity of Intestinal Enzymes in the Mucosa of Various Parts of the Intestine

In the absence of autoprobiotics, after the withdrawal of antibiotics (the group C1), only a tendency toward a decrease in the specific activity of maltase in mucosa of the upper intestine (duodenum, jejunum) was observed compared to the control group C0 (Figure 7A). With the introduction of indigenous bifidobacteria after the withdrawal of antibiotics (the group Bf), the specific activity of maltase in the mucosa of these parts of the intestine was restored to levels close to the control group C0, and there were no noticeable changes in this parameter in the ileum and colon compared to the control C0. Similar patterns were observed for the specific activity of alkaline phosphatase (APh) in the mucosa of various parts of the intestine in rats in groups C1 and Bf compared with the control group C0 (Figure 7B).

The specific activity of aminopeptidase N (APN) in the mucosa in group C1 (without autoprobiotics) changed insignificantly in the duodenum and jejunum, but was markedly increased (*p* < 0.05) in the colon compared with the control group C0 (Figure 7C). In contrast to the C1 group, in the Bf group (with indigenous bifidobacteria), the specific activity of APN in the mucosa was reduced in the duodenum compared to the C0 control group (*p* < 0.05), and it was close to the C0 control group in the colon.

### 3.6. The Activity of Intestinal Enzymes in the Chyme of the Colon

It is known that the large intestine is the site of degradation of digestive enzymes, including membrane intestinal enzymes, which enter it as part of the desquamated epithelium from the upper intestine [1]. Both pancreatic and bacterial proteases are involved in the degradation of digestive enzymes [1,3]. In this regard, a comparison of the activities of membrane digestive enzymes in the chyme fraction of the colon between groups C0, C1, and Bf made it possible to evaluate the features of the degradation of individual membrane enzymes in these groups.

Data presented on Figure 8A show that the specific activity of maltase in the chyme of the colon in the C1 group has a tendency towards a decrease compared control group C0. In contrast to the C1 group, the use of indigenous bifidobacteria in the Bf group resulted in a more significant decrease in specific activity of maltase in the chyme of the colon compared to the control group C0 (*p* < 0.05). Differences in this indicator between groups C1 and Bf were also noticeable (*p* < 0.01).

The specific activity of APh in the chyme of the colon in the group C1 decreased slightly compared to the control group C0, but in the group Bf it was significantly increased compared with group C1 (*p* < 0.01) and was characterized by a tendency to increase in comparison with the control group C0 (Figure 8B). The specific APN activity in the chyme of the colon in the C1 group tended to be higher compared with the C0 control group, and in the Bf group it was significantly higher than in the C0 control group (*p* < 0.05) (Figure 8C).

## 4. Discussion

It is generally accepted that probiotics, which most commonly include strains of lactobacilli and bifidobacteria, can improve intestinal microecology and, as a result, also have a positive effect on many metabolic and physiological processes in the body, in the regulation of which they are involved [4,6,7,8,10]. Despite a large number of works in this direction, the level of scientific confirmation of the effectiveness of specific probiotics remains insufficient. Even less is known about the effects of autoprobiotics, in which there has been a great deal of interest in recent years, on the digestive function and the gut microbiome.

In this work, we used a rat model of antibiotic-associated dysbiosis in order to evaluate the probiotic efficacy of indigenous bifidobacteria on microbiota and membrane enzymes involved in digestion and maintaining intestinal homeostasis.

Our results showed that the use of indigenous bifidobacteria (autoprobiotic) in rats after the withdrawal of antibiotics (the group Bf) accelerated the disappearance of dyspeptic symptoms of intestinal dysbiosis in them, did not change the weight gain of animals, and prevented an increase in chyme mass in the upper intestine, which occurred in the absence of autoprobiotics (the group C1). However, there were also undesirable consequences. In the Bf group and the C1 group, there was a decrease in the mass of the mucosa in the colon and an increase in the mass of the cecum including its contents compared with the control C0. But a more significant undesirable consequence was in the Bf group: The mass of the chyme in the colon significantly increased compared with the control group C0 and the C1 group. This effect may be due to the indirect action of indigenous bifidobacteria (by changing the content of some other bacteria or their metabolites) on the motor function of the colon. This assumption was supported, for example, by the fact that we found that in rats of the Bf group there was a correlation between the mass of the chyme in the colon and the content of bacteria of the genus *Blautia* and the family *Prevotella* in the chyme of this region (for the genus *Blautia* r = 0.52, *p* < 0.05, and for the *Prevotella* family r = −0.64, *p* < 0.05).

The results of the study of the microbiota in the feces of rats by qPCR showed a reduced content of opportunistic bacteria: *Proteus* spp. and enteropathogenic *Escherichia coli* in the Bf group compared to the group C1. Thus, these data confirm the effectiveness of indigenous bifidobacteria in suppressing opportunistic bacteria. Moreover, the Bf group showed an increased content of *Bifidobacterium* spp. compared to control group C0 and group C1. The latter fact was in good agreement with our data obtained by metagenomic analysis. In addition, metagenomic analysis revealed changes in the relative abundance of a number of beneficial bacteria in the group Bf compared to the control group C0 and the group C1. Thus, in group Bf, the relative abundance of bacteria in the genera *Ruminococcus, Prevotella*, and *Oscillospira* was reduced and in the genera *Bifidobacterium* spp., *Blautia* spp., and *Dorea* spp. was increased in comparison with the control group C0. The physiological significance of these changes on body systems remains unclear.

We also identified specific effects of indigenous bifidobacteria on the activities of three important intestinal enzymes: Maltase, alkaline phosphatase, and aminopeptidase N.

Maltase plays a key role in the breakdown of carbohydrates in adult mammals, participating in the final stages of hydrolysis of dietary glucose polymers, mainly starches [1]. In our study, we showed that in the case of the use of indigenous bifidobacteria after antibiotic withdrawal (the group Bf), the specific activity of maltase in the mucosa membrane of the upper intestine was noticeably higher than in group C1 (in the absence of autoprobiotic after antibiotic withdrawal) and was close to the level of this enzyme in the control group C0. Thus, one may assume that the improvement of the intestinal microbiota under the influence of indigenous bifidobacteria contributes to the restoration of the level of maltase activity in the mucosa of the upper intestinal tract. However, when studying the specific activity of maltase in the chyme of the colon in the group Bf, we found a significantly lower level of activity of this enzyme compared to the control group C0 and the group C1. Analyzing this result, it is important to keep in mind that the activity of the membrane enzyme in the chyme fraction of the colon reflects a dynamic balance between the rate of its entry into this region of the intestine (as a part of the desquamated epithelium from the overlying regions) and the rate of its degradation in this region with the participation of pancreatic and bacterial proteases. In addition, the rate of transit of the chyme in the colon has a significant effect on this process. Taking into account the slowed down rate of chyme transit in the colon in group Bf as compared to control group C0 and group C1 in our experiments, it was quite reasonable to expect a lower level of activity of the degrading enzyme.

Regarding the responses of the specific alkaline phosphatase activity in the intestinal mucosa to the introduction of an autoprobiotic or its absence after the withdrawal of antibiotics, in our experiments, basically the same regularities were observed as in the case of maltase. This is especially important due to the fact that APh, in addition to participating in the digestion of phosphoric acid esters and regulation of lipid absorption [1,17,18], also plays a key role in maintaining intestinal homeostasis, participating in the detoxification of the bacterial toxin LPS, controlling inflammation caused by LPS, and restricting the translocation of bacteria from the intestine to the lymphoid organs [19,20,21]. It is also important to note that under conditions of reduced chyme transit in the colon in the Bf group, the APh activity in this region was higher than in the C1 group (without autoprobiotic) and did not differ from the C0 control group. Thus, the presence of indigenous bifidobacteria after the withdrawal of antibiotics did not prevent the protective reaction from the intestinal APh.

APN, being a key enzyme in the final stages of hydrolysis of food proteins, is also involved in the degradation of biologically active peptides [22], in cholesterol transport, in immune responses, and can also serve as a receptor for antigens [20,23]. The use of indigenous bifidobacteria after the withdrawal of antibiotics in our experiments led to a decrease in the specific activity of APN in the mucosa of the duodenum in comparison with control group C0. At the same time, the specific activity of this enzyme did not change in the colon mucosa in the Bf group compared to the C0 control group, whereas it was increased in the C1 group. However, in the chyme fraction of the colon in the Bf group, as in the C1 group, the specific activity of APN was increased compared to the control C0. The physiological significance of the noted changes in the specific activity of AP-N in the mucosa of the duodenum and in the chyme of the colon in group Bf remains unclear. However, given the fact that inhibitors of peptidases, including AP-N, reduce colitis in mice [20,24], the APN reaction in the duodenum can be considered as positive, aimed to reduce the inflammatory process in the intestine.

## 5. Conclusions

In conclusion, the present study, using a rat dysbiosis model, showed that indigenous bifidobacteria can protect the microbiota and membrane enzymes, involved in digestion and maintaining homeostasis in the intestine, from disorders caused by dysbiosis; however, there are features in changing the listed parameters and an undesirable consequence in relation to the motor function of the colon which must be taken into account in the case of using such an autoprobiotic.

## Figures and Tables

**Figure 1 microorganisms-09-00522-f001:**
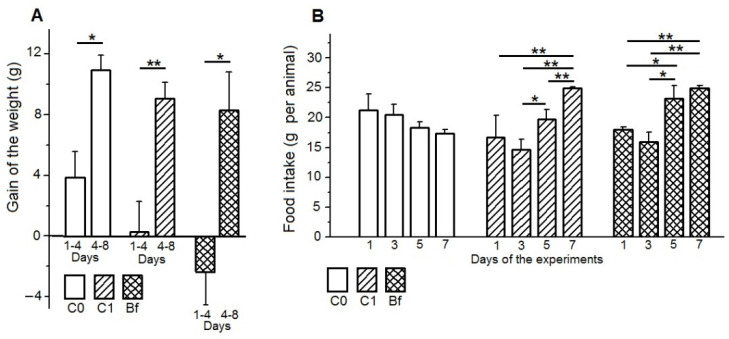
Influence of indigenous bifidobacteria on the gain of weight (**A**) and food intake (**B**) in a rat model of antibiotic-associated dysbiosis. (**A**,**B**) The rat groups: C0 (control)—without the introduction of antibiotics and autoprobiotics, C1—without the introduction of autoprobiotics after the withdrawal of antibiotics, and Bf—with the introduction of indigenous bifidobacteria after the withdrawal of antibiotics. (**A**) Student’s *t*-test, * *p* < 0.01; ** *p* < 0.003. (**B**) Student’s *t*-test, * *p* < 0.05, ** *p* < 0.003.

**Figure 2 microorganisms-09-00522-f002:**
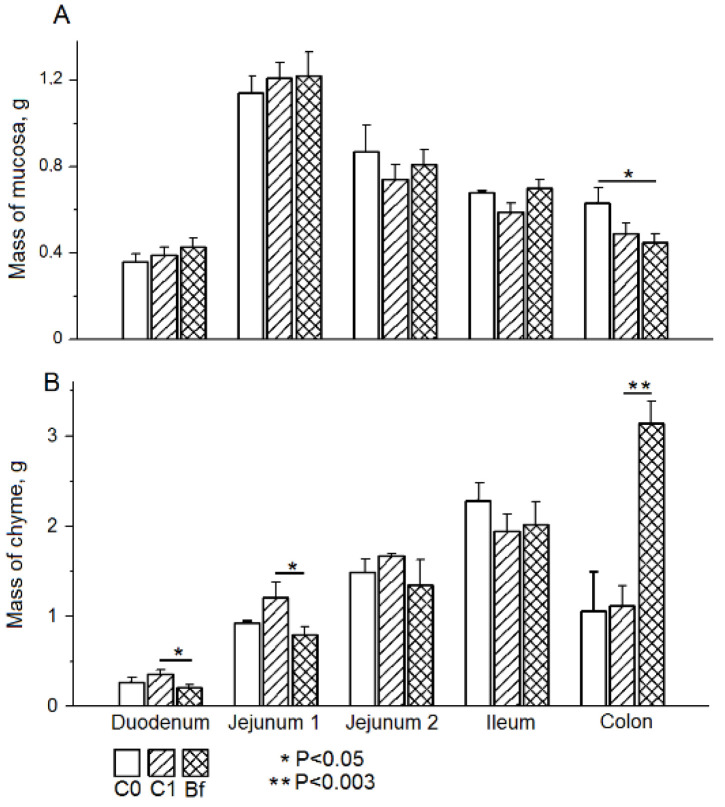
Influence of indigenous bifidobacteria on the mass of the mucosa (**A**) and chyme (**B**) in different parts of the intestine in a rat model of antibiotic-associated dysbiosis. (**A**,**B**) The rat groups: C0 (control)—without the introduction of antibiotics and autoprobiotics, C1—without the introduction of autoprobiotics after the withdrawal of antibiotics, and Bf—with the introduction of indigenous bifidobacteria after the withdrawal of antibiotics. (**A**) Student’s *t*-test, * *p* < 0.05. (**B**) Student’s *t*-test, * *p* < 0.05, ** *p* < 0.003.

**Figure 3 microorganisms-09-00522-f003:**
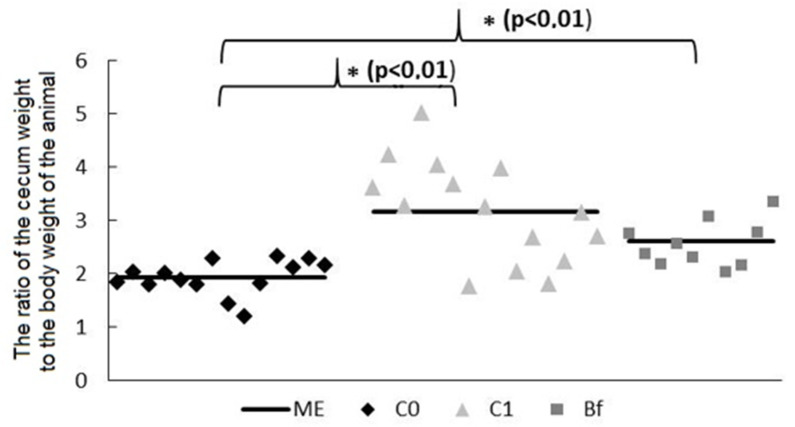
Changes in the ratio of the cecum weight (including to its contents) to the body weight of animal in a rat model of antibiotic-associated dysbiosis. The rat groups: C0 (control)—without the introduction of antibiotics and autoprobiotics, C1—without the introduction of autoprobiotics after the withdrawal of antibiotics, and Bf—with the introduction of indigenous bifidobacteria after the withdrawal of antibiotics. ME—median. Mann–Whitney U test, * *p* < 0.01.

**Figure 4 microorganisms-09-00522-f004:**
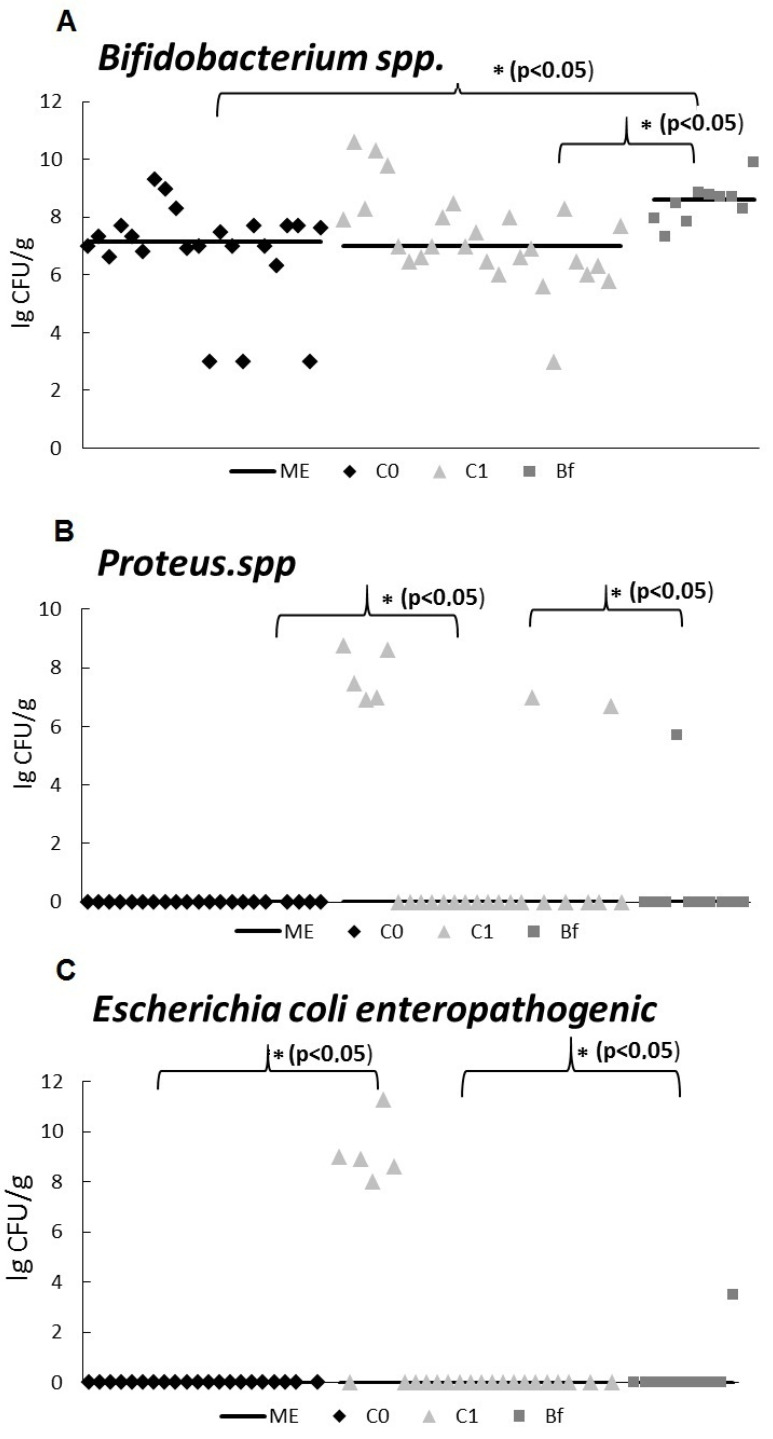
Quantitative content of marker bacteria in the fecal samples in a rat model of antibiotic-associated dysbiosis. The rat groups: C0 (control)—without the introduction of antibiotics and autoprobiotics, C1—without the introduction of autoprobiotics after the withdrawal of antibiotics, Bf—with the introduction of indigenous bifidobacteria after the withdrawal of antibiotics. (**A**–**C**) ME—median. Mann–Whitney U test, * *p* < 0.05.

**Figure 5 microorganisms-09-00522-f005:**
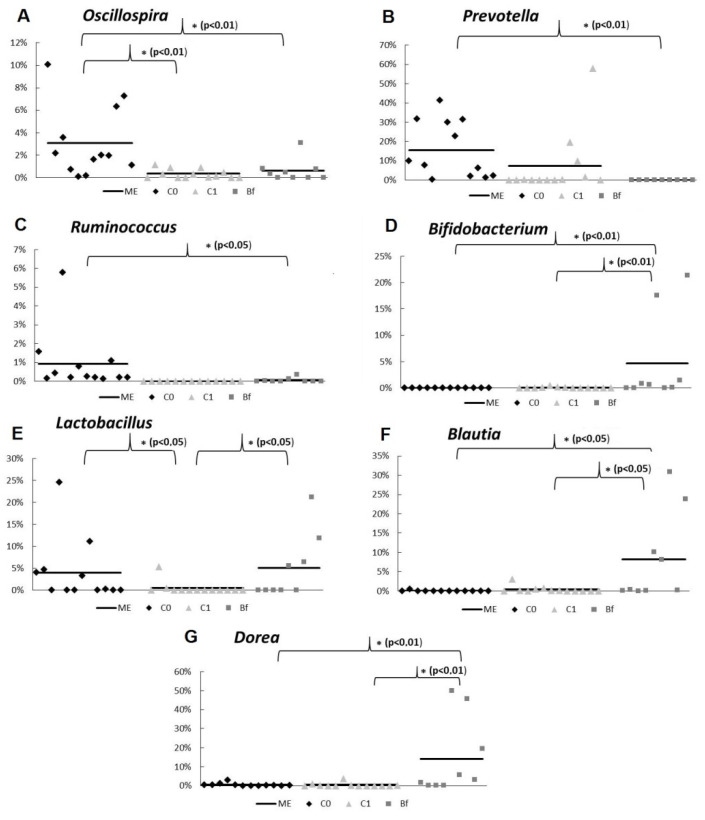
Statistically significant parameters of bacterial abundance in fecal samples of rats from different groups (metagenomic 16 S rRNA analysis, genus level). The rat groups: C0 (control)—without the introduction of antibiotics and autoprobiotics, C1—without the introduction of autoprobiotics after the withdrawal of antibiotics, and Bf—with the introduction of indigenous bifidobacteria after the withdrawal of antibiotics. (**A**–**G**) ME—median. Mann–Whitney U test, * *p* < 0.05.

**Figure 6 microorganisms-09-00522-f006:**
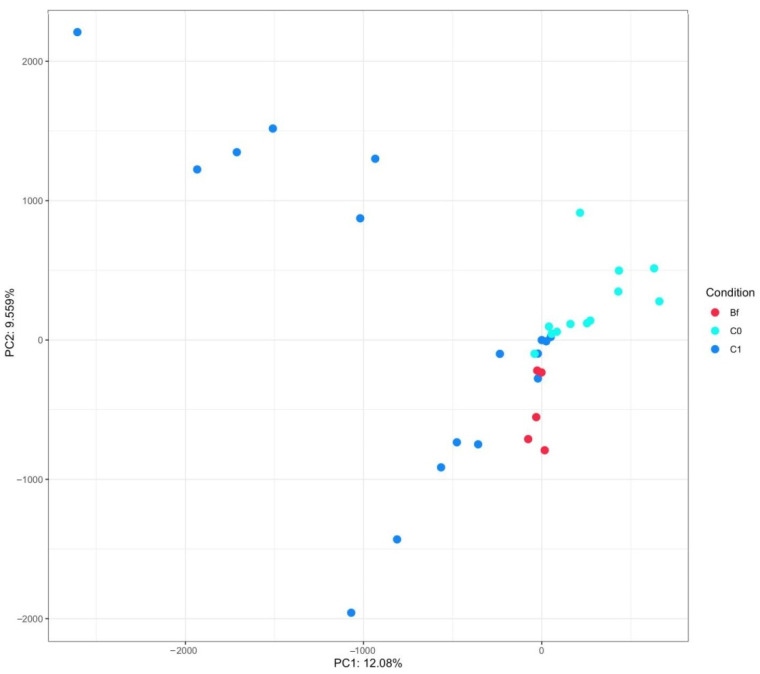
Principal component analysis fecal samples’ metagenome of rats from different groups.

**Figure 7 microorganisms-09-00522-f007:**
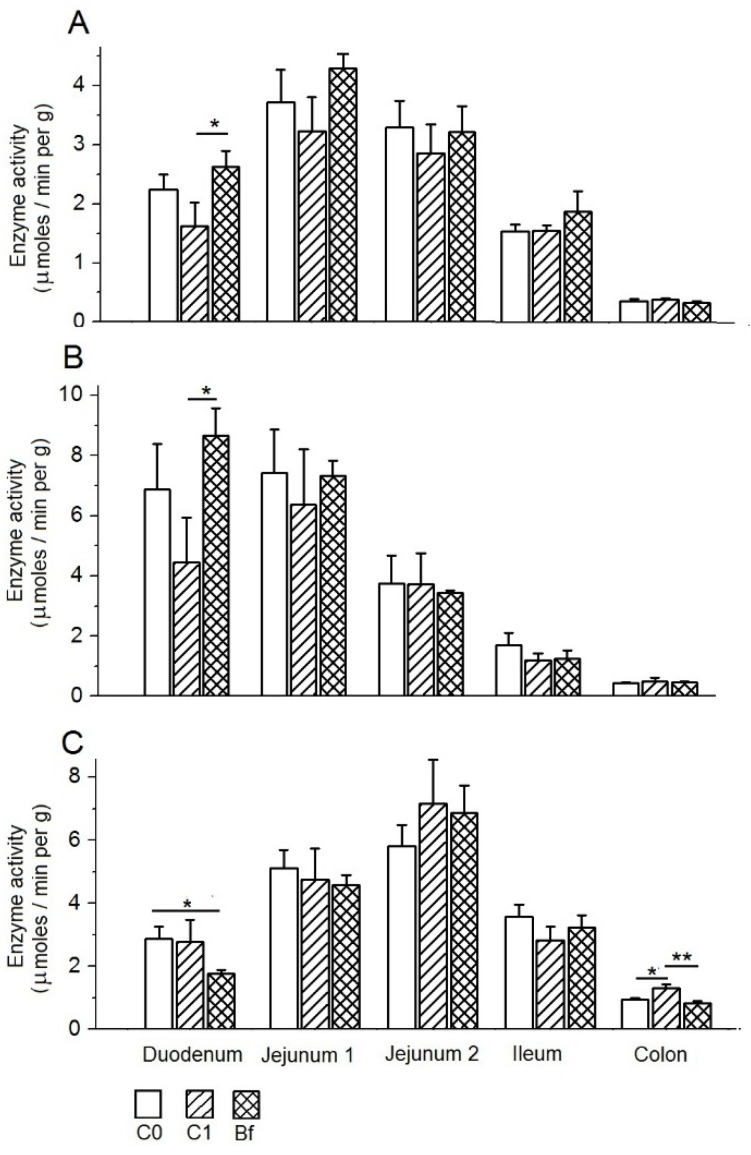
Effects of indigenous bifidobacteria on the specific activities of maltase (**A**), alkaline phosphatase (**B**), and aminopeptidase N (**C**) in the mucosa of different parts of the intestine in a rat model of antibiotic-associated dysbiosis. (**A**–**C**) The rat groups: C0 (control)—without the introduction of antibiotics and autoprobiotics, C1—without the introduction of autoprobiotics after the withdrawal of antibiotics, and Bf—with the introduction of indigenous bifidobacteria after the withdrawal of antibiotics. (**A**,**B**) Student’s *t*-test, * *p* < 0.05. (**C**) Student’s *t*-test. * *p* < 0.02, ** *p* < 0.01.

**Figure 8 microorganisms-09-00522-f008:**
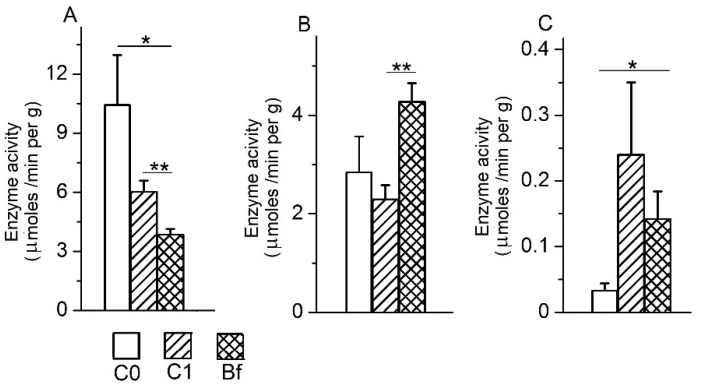
Effects of indigenous bifidobacteria on the specific activities of maltase (**A**), alkaline phosphatase (**B**), and aminopeptidase N (**C**) in the chyme of different parts of the intestine in a rat model of antibiotic-associated dysbiosis. (**A**–**C**) The rat groups: C0 (control)—without the introduction of antibiotics and autoprobiotic, C1—without the introduction of autoprobiotic after the withdrawal of antibiotics, and Bf—with the introduction of indigenous bifidobacteria after the withdrawal of antibiotics. (**A**) Student’s *t*-test, * *p* < 0.05, ** *p* < 0.01. (**B**) Student’s *t*-test * *p* < 0.01. (**C**) Students *t*-test, * *p* < 0.02, ** *p* < 0.05.

**Table 1 microorganisms-09-00522-t001:** DNA primers used for the *Bifidobacterium* spp. identification.

Forward	5′gcgtgcttaacacatgcaagtc3′
Reverse	5′cacccgtttccaggagctatt3′
Oligonucleotide sequences of the TaqMan probe	5′tcacgcattactcacccgttcgcc3′

**Table 2 microorganisms-09-00522-t002:** Experimental design.

Groups	Treatment (1–3 Days)	Treatment (4–8 Days)	Analysis of Samples
**C0**	Distilled water	PBS	Fecal samples were harvested 7 days before the start of administration of antibiotics for indigenous bifidobacteria strains’ isolation and preparation of autoprobiotic. Fecal samples harvested on days 0 and 9 of experiments were used for microbiota study.
**C1**	Ampicillin + metronidazole	PBS	Fecal samples harvested on days 0 and 9 of experiments were used for microbiota study. Samples of mucosa and chyme were taken from various parts of the intestine on ninth day of experiments for analysis of the activities of digestive enzymes.
**Bf**	Ampicillin + metronidazole	Autoprobiotic *Bifidobacterium* spp.	

**Table 3 microorganisms-09-00522-t003:** DNA primers used in this study for the metagenome analysis.

V3-V4 16S Region Sequencing Primers V4 16S Region Sequencing Primers		Amplicon Size bp
Forward (341)	tcgtcggcagcgtcagatgtgtataagagacagcctacgggnggcwgcag	464
Reverse (785)	gtctcgtgggctcggagatgtgtataagagacaggactachvgggtatctaatcc	

## Data Availability

Data available in a publicly accessible repository.

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
