# Peer review of "Gut Digestive Function and Microbiome after Correction of Experimental Dysbiosis in Rats by Indigenous Bifidobacteria"

_microorganisms, 2021, doi:10.3390/microorganisms9030522_

Round 1

Reviewer 1 Report

The study focused on determining the efficacy of auto-probiotic supplementation against gut dysbiosis. The study concept is very interesting and within the scope of the journal. However, the manuscript has a lot of opportunities for improvement. Specifically, the metagenomic data could have been more focused, authors could carry out a deep comparison between the fecal metagenomes with/without the treatment (suggest exploring alpha- and beta- diversity indices, taxa richness and evenness, homogeneity of multivariate dispersions test, comparison between taxa specific normalized read counts, etc). The most laborious/expensive part of the metagenomic studies is sample processing and sequencing, the authors have already completed that part. Therefore, the suggested analyses (of the sequencing data) will require minimal resources/input but will add significant value to the manuscript. The following comments should be addressed.      

Line 28: “We concluded that indigenous bifidobacteria can protect the microbiota and intestinal digestive enzymes in the intestine from disorders caused by dysbiosis…”. A negative control potentially using a non-indigenous bifidobacteria would add significant value to the study. Without this negative control, there is not enough evidence in this manuscript to make the above statement on biological role of “indigenous” bifidobacteira on digestive disorders.

Lines 90-95: How did the authors determine culture purity of the indigenous strain? Was it a pure culture of one strain or a missed culture? Methods and justifications need to be clarified here.

Line 101: please clarify.

Line 104: Why PBS was used in the control group? Was the experimental group fed auto-probiotic suspension in PBS? Again, a non-indigenous probiotic control and a comparison between indigenous vs non-indigenous probiotic would be valuable.

Lines 137-143: Need to add sequencing parameters, e.g., read length, paired/unpaired, detailed library prep protocol, number of samples run in a flow cell, etc.

Lines 145-152: Need details of the metagenomic analyses, e.g., all software used with versions, software parameters (if not default), quality trimming, etc. In addition, how the read numbers were normalized?

Lines 153-160: Why did the authors focus specifically on maltase, phosphatase, and peptidase activities?

Lines 170-172: Need to add numerical values with relevant statistics for these results.

Figure 3: lg CFA/g?

Author Response

See attachment file Response to reviewer 1

Reviewer 2 Report

[General comments]

This article describes the influence of administered bifidobacteria, which were isolated from the self-intestine, on the gut microbiota and digestive enzymes. The idea to use the beneficial bacteria residing in the intestine of ourselves for restoring from antibiotic-induced dysbiosis, is quite interesting. Right now, fecal microbiota transplantation has been confirmed to be effective to treat Clostridioides difficile-associated diarrhea, but its safety is still beyond control. Therefore, the usage of isolated bacteria seems to be safer than that of whole fecal material. In this sense, this study is highly promising. However, there are many issues to be addressed until this study will be completed.

[Specific comments]

  1. Lanes 89-95: The authors describe that Bifidobacterium has been isolated from stool samples of healthy rat. However, there are little explanations how they have isolated the bacterium in which isolation method, how they have identified the bacteria as Bifidobacteria (probably based on 16S rRNA sequence), and how many species of Bifidobacteria have been administered to rats. We know that the Bifidobacteria used here have been isolated from the stool, but is that all to confirm that the bacterium is truly indigenous?
  2. Lanes 102-105: I think that “C1 group received antibiotics and PBS”.
  3. Lane 117: What is AC content?
  4. Lanes 168-176: This section describes the most important results in this study, but there is no figure/table. Data on body weight, extent of diarrhea or constipation, and fecal consistency should be presented for the readers to understand the results more easily.
  5. Figure 1: This figure shows the mass of chyme in three groups. I imagine that the mass of chyme may be affected by the amount of food intake. To consider the significance of the change in chyme mass, data on the amount of food intake are necessary.
  6. Lanes 207-227: The change in gut microbiota is observed by giving Bifidobacteria, and the level of Proteus and enteropathogenic E. coli is individually variable in C1 group. Are the number of these bacteria correlated with the extent of diarrhea or constipation?
  7. Lanes 237-259: Digestive enzymes of mucosa are affected by ingesting Bifidobacteria. The influence of Bifidobacteria administration seems to be different from the site of intestine. Do the authors have good idea to explain how administered Bifidobacteria affects digestive enzymes differently dependent on the intestinal sites?
  8. The number of figures in the text and the figure legends is not matching in several places. Please check carefully.
  9. Discussion is too long and redundant (Lanes 333-379 and Lanes 423-469 are the same sentences).

Author Response

See attachment file Response to reviewer 2

Round 2

Reviewer 1 Report

Section 2.5.1: Authors should add read length for metagenomic sequencing.

Section 2.5.2: Authors must submit their raw sequences to a public domain and add accession numbers for their submission in this section. It is common practice for genomic and metagenomic studies to make sequencing data publicly available. In addition, authors need to explain how the reads were quality filtered, how many raw reads were obtained and how many were retained after quality filtering, etc. Without these information, metagenomic section seems incomplete. I would suggest the authors to follow relevant literature for presenting the metagenomic part of the study (in methods, results, and discussion sections).

Author Response

Comments and Suggestions for Authors

Section 2.5.1: Authors should add read length for metagenomic sequencing.

Section 2.5.2: Authors must submit their raw sequences to a public domain and add accession numbers for their submission in this section. It is common practice for genomic and metagenomic studies to make sequencing data publicly available. In addition, authors need to explain how the reads were quality filtered, how many raw reads were obtained and how many were retained after quality filtering, etc. Without these information, metagenomic section seems incomplete. I would suggest the authors to follow relevant literature for presenting the metagenomic part of the study (in methods, results, and discussion sections).

Answer:

Section 2.5.1: The read length for metagenomic sequencing is shown in table. 3 (line: 80).

Section 2.5.2: The remark has been taken into account. Section 2.5.2 (line: 99) now contains information on filtering quality and how many raw reads were received and how many were retained after quality filtering. Detailed results and conditions of sequencing are presented in the Supplementary Materials section.

Reviewer 2 Report

I have confirmed that the authors have revised finely and sincerely according to my comments. It is much better to identify the species of Bifidobacteria administered, to make this method more standard. The revised version will give useful information to the both basic and applied researches.

Author Response

Comments and Suggestions for Authors

I have confirmed that the authors have revised finely and sincerely according to my comments. It is much better to identify the species of Bifidobacteria administered, to make this method more standard. The revised version will give useful information to the both basic and applied researches.

Answer:   Unfortunately, in this work, we did not set ourselves the task of identifying the types of introduced bifidobacteria and analyzed only a possibility of using such bacterial genus to restore the microbiota and function of the gastrointestinal tract under the conditions of dysbiotic action of antibiotics. We are agree completely with you that knowledge of the composition of autoprobiotic bifidobacteria in our study might be useful for further fundamental and applied researches. In future, we are going to continue the researches in this direction.